# Assessment of adherence to the World Health Organization's prescribing indicators at the family medicine clinic of a quaternary facility

Nana Ama Buadiba Osei[1]*, Daniel Freeman Owusu Ansah[1], Kofi Bedu Tetteh[2], Angela Appiah Kuffuor[1], Fareeda Serwaa Brobbey[1], Bismark Sarfo[3]

**1** Pharmacy Directorate, University of Ghana Medical Centre, Legon, Greater Accra, Ghana, **2** Pharmacy Department, Trust Hospital Limited, Osu, Greater Accra, Ghana, **3** School of Public Health, University of Ghana, Legon, Greater Accra, Ghana

☯ These authors contributed equally to this work.

\* namaosei@yahoo.com

## Abstract

To promote rational drug use in developing countries, it is important to assess drug use patterns. This study assessed the drug prescription patterns of the family medicine clinic at the Outpatient Pharmacy of the University of Ghana Medical Centre using the World Health Organization's drug use indicators. An analytic, cross-sectional survey with data extracted from patient's electronic medical records was carried out. Questionnaires were given to all prescribers in the family medicine clinic to evaluate factors related to rational medicine use. Frequencies and percentages were employed for description with further analysis, including Zero-inflated Poisson regression and logistic regression, used to determine associations between variables with a 95% confidence interval. Of the 600 participants whose prescriptions were analyzed, 367 (61.17%) were male and 233 (38.83%) were female. The prescribers interviewed were 3 males and 7 females. The mean number of medications per prescription was 1.4 (SD = 1.61), with antibiotics and injections making up 12.74% (n = 107) and 4.17% (n = 35) respectively. Generic prescriptions were 34.88% (n = 293) and those from the Essential Medicines List (EML) were 72.38% (n = 608). Prescriptions with a record of diagnosis were 50.83% (n = 305). Patients with comorbidities were shown to have a 52.2% lower prevalence rate of the total number of medications prescribed compared to those without comorbidities (p-value <0.001). Female patients have 46.4% reduced odds of being prescribed an antibiotic compared to male patients (p-value 0.012). The odds of a patient with a chronic condition being prescribed an antibiotic is 93.2% more than that of a patient without a chronic condition (p-value = 0.025). Additionally, the prevalence of drugs prescribed from the EML for a patient with a chronic condition is 74.4% lower than that prescribed for patients without a chronic condition (p-value = 0.048). There was moderate adherence to rational prescribing. Three prescribing indicators met reference standards, these

**Data availability statement:** All relevant data are within the manuscript.

**Funding:** The author(s) received no specific funding for this work.

**Competing interests:** The authors have declared that no competing interests exist.

were: average number of medicines per encounter, percentage of prescriptions with an injectable and percentage of encounters with antibiotics. Rational drug prescribing may be enhanced through training, guidelines, EML distribution, drug and therapeutics committee support and integrated Clinical Decision Support Systems (CDSS).

## Introduction

Drugs are essential in achieving good clinical outcomes in patients [1]. Therefore, rational use of medicines should be well integrated into any health system to preserve the integrity and quality of drugs provided to patients. The World Health Organization (WHO) defines rational use of medicines (RUM) as patients "*receiving medications that are appropriate for their clinical needs, administered in the right dosage to meet their individual needs, for enough time, and at the most affordable price for both the patient and their community*" [2]. Adherence to good prescribing practices is essential to achieve quality healthcare delivery. For this reason, the World Health Organization (WHO) has developed quality indicators for the evaluation of prescribing practices to prevent any negative consequences associated with irrational medicine use such as poor treatment outcomes, higher treatment costs and increased risk of adverse reactions [3]. Despite the efforts made by the WHO to ensure medicines are used rationally, more than 50% of prescribed drugs worldwide are utilized inappropriately and more than 50% of patients do not take their medicines as they are told to [4]. Again, about 33% of the population worldwide do not have access to essential medicines which is concerning [5]. Drug utilization studies are therefore used periodically to ensure RUM. These studies aid in the evaluation of the quality of drugs and make clinical care cost-effective [6]. Drug utilization studies involve the review of data from prescriptions. Prescriptions reflect the physician's knowledge and how well they utilize medications in the treatment of patients [1]. Due to a myriad of factors such as lack of qualified workers, gaps in knowledge and lack of funds, underdeveloped countries continue to face challenges regarding RUM, which is of public health concern as it affects clinical outcomes in patients [7]. In a developing country like Ghana, there is a need to understand prescription patterns in order to maximize the use of the limited resources provided [6]. Prescribing practices have been assessed in many studies worldwide, highlighting different results of irrational prescribing. A 2010 study in Saudi Arabia showed that an average of 2.4 drugs were prescribed per encounter, with 61.2% prescribed by generic name and 32.2% involving antibiotics [8]. Similarly, on average, 3.4 drugs were prescribed per encounter, with 71.6% being generic prescriptions, and antibiotics and injections being involved in 48.9% and 27.1% of encounters, respectively in a study conducted in Pakistan [9]. A 2021 study in Ghana showed that on average, 3 medicines were prescribed per encounter with a patient, with generics accounting for 76% of prescriptions, and 7% of encounters including injections [10]. A further study in Northern Ghana reported an average of 3.9 medicines prescribed per encounter, with 55% and 14% of encounters involving antibiotics and injections, respectively, while prescribing by generic name was observed in 53%

of cases [11]. These studies highlight areas that can be improved to enhance prescribing practices. It is therefore imperative for facilities to regularly assess prescribing practices. Results from these studies could be used to sensitize prescribers and draw attention to inappropriate medication use. Such studies are beneficial to policymakers in terms of reviewing procurement and prescribing policies within the facility. Given that University of Ghana Medical Centre, Legon (UGMC) is a relatively new healthcare facility with personnel from different backgrounds, monitoring prescription patterns is essential to the implementation of appropriate measures to reduce potential medication errors. This study assessed prescribing practices at UGMC using the WHO prescribing indicators. It was conducted to contribute to the existing body of knowledge and aid in identifying areas for improvement in prescribing practices at this quaternary healthcare facility [12].

## Methods

### Study design and setting

An analytic cross-sectional survey was conducted at UGMC to evaluate the WHO core prescribing indicators using prescription data from patient medical records spanning January 2022 to May 2023. This retrospective data collection followed the WHO Manual's guidelines on investigating drug use in health facilities [11]. Additionally, in August 2023, physicians at the Family Medicine Clinic were engaged through a questionnaire to gather further insights into prescribing practices. The study was conducted at the University of Ghana Medical Centre, Legon. UGMC is a quaternary government healthcare facility located in the southernmost part of the University of Ghana, Legon campus. This referral center has a 1000-bed capacity and began operations in 2018. Medications in this institution are sourced through a decentralized procurement system following the Ministry of Health's Medicines Procurement Framework Agreement [13] alongside purchases with petty cash in emergencies. The outpatient department of the Centre offers a wide range of specialist services.

### Sample size determination

The total number of prescriptions in the medical records of patients at the family medicine clinic at UGMC from January 1, 2022, to May 31, 2023, was 23,690. In adherence to the WHO guidelines, a minimum sample size of 600 is required for an RUM survey [14,15]. A sample frame was established in Excel to chronologically order the study population based on prescriptions within this period. STATA software was then employed for the random selection of medical records, [14,15] after which 600 prescriptions from the total number of prescriptions were selected. Irrational prescribing was assessed from the selected prescriptions using the WHO prescribing indicators.

### Data collection tool and approach

The outcomes of interest encompassed the prevalence of polypharmacy, commonly defined as taking five or more regularly prescribed medications, the prevalence of generic name-based prescribing, the prevalence of antibiotic prescriptions, the prevalence of injection prescriptions, and adherence to the essential drug list during the prescribing process. Patients of different age groups, genders, and medical conditions were included, as the family medicine clinic caters to a diverse range of patients. Prescriptions with a combination of medicines were counted as one. Prescriptions that did not contain all the information needed, such as the names of the drugs, and their dosage forms, and those that had only medical equipment and supplies, were excluded from the study. Questionnaires were administered to all physicians who were actively involved in prescribing medications to patients at the family medicine clinic. Written consent was sought from the participants before the questionnaire's administration.

### Data quality control

To ensure consistency in results, all data collectors were trained and allowed to extract the data together at the study site. The data collected was then checked to ensure that all information required is recorded before entry into Microsoft Excel. The data was rechecked to ensure correctness in order to produce reliable results.

## Statistical analysis

To assess the performance of UGMC in terms of drug utilization, an index system developed by Zhang and Zhi [16] was adopted in this study. An index was determined for each prescribing indicator using a formula derived from previous studies. To calculate the absence of polypharmacy, rational antibiotic and injection safety indices, the formula below was used:

$$\textbf{Index value} = \frac{Optimal\ value}{Observed\ value}$$

Whereas generic prescribing, prescribing from the essential medicines list and recording of diagnosis were determined by the following formula:

$$\textbf{Index value} = \frac{Observed\ value}{Optimal\ Value}$$

Each prescribing indicator had an optimal index set at 1 thus any observed value close to 1 was ideal. The maximum value for this study was 6 which was obtained by adding up all 6 indices.

The values computed for each indicator were compared with the optimal level prescribed by the WHO. Simple descriptive statistics were conducted for categorical variables. Means and standard deviations were determined for continuous variables. Frequencies and percentages through tables and charts were used to report results. Further analysis including the Zero-inflated Poisson regression and logistic regression were used to assess the magnitude of the association between identified variables. This regression analysis of all the factors was assessed through the adjusted and unadjusted models to establish which of the factors were driving the relationships in the variables. Both the crude and adjusted odds ratios and incidence risk ratios were estimated with their corresponding 95% confidence interval. Strict privacy and confidentiality measures were taken during the data extraction process from medical records. Patient identifiers, including names, addresses, and telephone numbers, were either removed or anonymized to safeguard privacy. The data was securely stored, with access restricted solely to authorized research personnel, ensuring strict confidentiality, and preventing unauthorized access or disclosure.

## Ethical considerations

The research protocol, including the data extraction process, was submitted to the UGMC institutional review board (IRB) for review and approval. Approval was secured before data extraction commenced (Protocol number: UGMC-IRB/ MSRC/042/2023) and informed written consent was obtained from the prescribers before filling out the questionnaires.

## Results

### Demographic characteristics of patients

A total of 600 prescriptions were assessed in this study. Almost two-thirds of the patients (367, 61.17%) were males. More than a quarter of the patients (156, 26.00%) were between the ages of 31–40 years (Table 1). The mean age of the patients was 43.82(SD = 17.88 years).

### Prescribing indicators

The observed values of the prescribing indicators along with the index of rational drug prescribing (IRDP) are shown in Table 2. The percentage of recording of diagnosis was 50.83% with an IRDP of 0.51, the average number of medicines per visit was 1.4 with an IRDP of 1 and the percentage of medicines prescribed that were in the essential medicines list for Ghana was 72.38% with an IRDP of 0.72.

**Table 1. Demographic characteristics of patients.**

| Variable | Number | Percentage |
|---|---|---|
| **Sex** | | |
| Male | 367 | 61.17 |
| Female | 233 | 38.83 |
| **Age (years)** | | |
| ≤ 10 | 3 | 0.50 |
| 11–20 | 24 | 4.00 |
| 21–30 | 138 | 23.00 |
| 31–40 | 156 | 26.00 |
| 41–50 | 81 | 13.50 |
| 51–60 | 71 | 11.83 |
| 61–70 | 65 | 10.83 |
| 71–80 | 47 | 7.83 |
| > 80 | 15 | 2.50 |
| **Presence of Chronic condition** | | |
| Present | 136 | 22.67 |
| Absent | 464 | 77.33 |
| **Presence of comorbidity** | | |
| Present | 43 | 7.17 |
| Absent | 557 | 92.83 |

## Observed and optimum IRDP

The observed IRDP for polypharmacy, prescribing injectables and antibiotics was 1 each which is the same as the optimum IRDP defined by the WHO/INRUD. The overall observed IRDP of 4.58 was below the optimum IRDP of 6 indicating poor drug prescribing outside WHO standards (Table 2).

## Poisson regression of demographic characteristics and number of drugs prescribed

Table 3 shows the detailed association of the number of drugs prescribed for a patient during a visit to the hospital using the Poisson regression. For the univariate analysis, the presence of comorbidities ($p < 0.001$) and a chronic condition ($p < 0.001$) was significantly associated with the number of drugs prescribed for a patient during a visit to the hospital. Patients with chronic conditions have a 55.8% lower prevalence rate of the total number of medications prescribed compared to those without chronic conditions. Patients with comorbidities have a 52.2% lower prevalence rate of the total number of medications prescribed compared to those without comorbidities. For the multivariate analysis, while holding all other variables in the model constant, the presence of comorbidity ($p < 0.001$) was significantly associated with the number of drugs prescribed for a patient during a visit to the hospital. The prevalence rate of the total number of drugs prescribed for a patient with a chronic condition was 58.5% lower than that of prescribed for patients without chronic conditions.

## Zero-inflated Poisson regression of demographic characteristics and number of drugs prescribed in their generic names

Table 4 shows the detailed association of the number of drugs prescribed in their generic names for a patient during a visit to the hospital using the zero-inflated Poisson regression. For the univariate analysis, none of the independent variables was significantly associated with the number of drugs prescribed with their generic names. For the multivariate analysis,

**Table 2. Observed values of prescribing indicators with their IRDP.**

| Prescribing indicators | Observed values | WHO Standards | Index of RDP (IRDP) |
|---|---|---|---|
| Records of diagnosis | 50.83% | 100% | 0.51 |
| Average number of drugs per visit | 1.4 | < 2 | 1.00 |
| Drugs prescribed with generic names | 34.88% | 100% | 0.35 |
| Patients with injection(s) prescribed | 4.17% | < 25% | 1.00 |
| Patients with antibiotic(s) prescribed | 12.74% | < 30% | 1.00 |
| Drugs prescribed from the Essential Medicines List | 72.38% | 100% | 0.72 |
| Overall IRDP | | | 4.58 |

**Table 3. Poisson regression of demographic characteristics and number of drugs prescribed.**

| Variable | Unadjusted | | Adjusted | |
|---|---|---|---|---|
| | IRR (95% CI) | p-value | IRR (95% CI) | p-value |
| **Sex** | | | | |
| Male | Ref | | Ref | |
| Female | −0.077 (−0.216–0.063) | 0.282 | −0.100 (−0.241–0.040) | 0.161 |
| **Age** | 0.001 (−0.003–0.005) | 0.607 | −0.003 (−0.007–0.001) | 0.150 |
| **Presence of chronic condition** | | | | |
| Absent | Ref | | Ref | |
| Present | 0.442 (0.296–0.587) | < 0.001* | 0.415 (0.240–0.590) | < 0.001* |
| **Presence of comorbidity** | | | | |
| Absent | Ref | | Ref | |
| Present | 0.478 (0.262–0.693) | < 0.001* | 0.207 (−0.050–0.463) | 0.114 |

*IRR means Incidence Risk Ratio, Ref means Reference group.

while holding all other variables in the model constant, none of the independent variables was significantly associated with the number of drugs prescribed with their generic names.

## Logistic regression of demographic characteristics and the presence of an antibiotic in a prescription

Table 5 shows the detailed association of the demographic characteristics and antibiotics prescribed for a patient during a visit to the hospital using logistic regression. For the univariate analysis, sex (p = 0.013) was significantly associated with antibiotics prescribed for a patient visiting the hospital. For the multivariate analysis, while holding all other variables in the model constant, sex (p = 0.012) and the presence of chronic condition (p = 0.025) were significantly associated with the presence of an antibiotic in a prescription. A female patient had 46.4% reduced odds of being prescribed an antibiotic compared to a male patient. The odds of a patient with a chronic condition being prescribed an antibiotic was 93.2% more than that of a patient without a chronic condition when confounders are controlled.

## Logistic regression of demographic characteristics and the presence of an injection in a prescription

Table 6 shows the detailed association of injections prescribed for a patient during a visit to the hospital using logistic regression. For the univariate analysis, none of the independent variables was significantly associated with the number of injectables prescribed. For the multivariate analysis, while holding all other variables in the model constant, none of the independent variables was significantly associated with the number of injectables prescribed.

**Table 4. Zero-inflated Poisson regression of demographic characteristics and number of drugs prescribed with generic names.**

| Variable | Unadjusted | | Adjusted | |
|---|---|---|---|---|
| | IRR (95% CI) | p-value | IRR (95% CI) | p-value |
| **Sex** | | | | |
| Male | Ref | | Ref | |
| Female | 0.128 (−0.227–0.483) | 0.479 | −0.060 (−0.340–0.220) | 0.675 |
| **Age** | 0.006 (−0.003–0.015) | 0.220 | −0.001 (−0.009–0.008) | 0.901 |
| **Presence of chronic condition** | | | | |
| Absent | Ref | | Ref | |
| Present | 0.219 (−0.151–0.588) | 0.246 | 0.288 (−0.154–0.729) | 0.202 |
| **Presence of comorbidity** | | | | |
| Absent | Ref | | Ref | |
| Present | 0.058 (−0.484–0.601) | 0.887 | −0.141 (−0.773–0.490) | 0.661 |

*IRR means Incidence Risk Ratio, Ref means Reference group.

**Table 5. Logistic regression of demographic characteristics and prescription of antibiotics.**

| Variable | Unadjusted | | Adjusted | |
|---|---|---|---|---|
| | OR (95% CI) | p-value | OR (95% CI) | p-value |
| **Sex** | | | | |
| Male | Ref | | Ref | |
| Female | 0.542 (0.334–0.881) | 0.013* | 0.536 (0.329–0.874) | 0.012* |
| **Age** | 0.992 (0.980–1.005) | 0.236 | 0.989 (0.976–1.003) | 0.127 |
| **Presence of chronic condition** | | | | |
| Absent | Ref | | Ref | |
| Present | 1.442 (0.880–2.363) | 0.146 | 1.932 (1.088–3.428) | 0.025* |
| **Presence of comorbidity** | | | | |
| Absent | Ref | | Ref | |
| Present | 0.853 (0.350–2.080) | 0.726 | 0.643 (0.235–1.758) | 0.390 |

*OR means Odds Ratio, Ref means reference group.

## Zero-inflated Poisson regression of demographic characteristics and number of drugs prescribed from the Essential Medicines List of Ghana

Table 7 shows the detailed association of the number of drugs prescribed from the essential medicines list (EML) for a patient during a visit to the hospital using the zero-inflated Poisson regression. For the univariate analysis, none of the independent variables was significantly associated with the number of drugs prescribed from the EML. For the multivariate analysis, while holding all other variables in the model constant, the presence of chronic condition (p = 0.048) was significantly associated with the number of drugs prescribed from the EML. The findings indicate that the prevalence of drugs prescribed from the EML for a patient with a chronic condition was 74.4% lower than that prescribed for patients without chronic conditions when confounders were controlled.

## Demographic characteristics of prescribers

Ten prescribers participated in this study. Most of the prescribers (7, 70.00%) were females. The median age group of the prescribers was 31–35 years. The majority of the prescribers (9, 90.00%) were medical officers with 1 specialist (10.00%) (Table 8).

**Table 6. Logistic regression of demographic characteristics and prescription of injectables.**

| Variable | Unadjusted | | Adjusted | |
|---|---|---|---|---|
| | OR (95% CI) | p-value | OR (95% CI) | p-value |
| **Sex** | | | | |
| Male | Ref | | Ref | |
| Female | 1.490 (0.687–3.229) | 0.312 | 1.500 (0.691–3.259) | 0.305 |
| **Age** | 0.993 (0.971–1.015) | 0.559 | 0.992 (0.969–1.016) | 0.528 |
| **Presence of chronic condition** | | | | |
| Absent | Ref | | Ref | |
| Present | 0.974 (0.385–2.463) | 0.955 | 0.997 (0.328–3.027) | 0.996 |
| **Presence of comorbidity** | | | | |
| Absent | Ref | | Ref | |
| Present | 1.038 (0.238–4.537) | 0.960 | 1.135 (0.195–6.587) | 0.888 |

*OR means Odds Ratio, Ref means reference group.

**Table 7. Zero-inflated Poisson regression of demographic characteristics and number of drugs prescribed found on the EML.**

| Variable | Unadjusted | | Adjusted | |
|---|---|---|---|---|
| | IRR (95% CI) | p-value | IRR (95% CI) | p-value |
| **Sex** | | | | |
| Male | Ref | | Ref | |
| Female | −0.090 (−0.326–0.052) | 0.157 | −0.085 (−0.307–0.137) | 0.453 |
| **Age** | −0.001 (−0.006–0.004) | 0.694 | −0.003 (−0.008–0.003) | 0.146 |
| **Presence of chronic condition** | | | | |
| Absent | Ref | | Ref | |
| Present | 0.203 (−0.017–0.422) | 0.070 | 0.256 (0.002–0.511) | 0.048* |
| **Presence of comorbidity** | | | | |
| Absent | Ref | | Ref | |
| Present | 0.067 (−0.269–0.403) | 0.696 | −0.065 (−0.454–0.323) | 0.637 |

*IRR means Incidence Risk Ratio, Ref means Reference group.

## Knowledge and training on RUM

Out of the 10 prescribers surveyed, 7 (70.0%) had heard of the Rational Use of Medicines (RUM) standards. Half (50.0%) indicated that, according to these standards, no more than four medicines should be prescribed per patient encounter. Additionally, 80.0% stated that medications should be prescribed using their generic names. Regarding antibiotic prescribing, 30.0% reported that RUM standards recommend no more than two antibiotics per patient encounter, while another 30.0% believed the limit was four antibiotics. Only 20.0% of prescribers had received training on RUM. Those trained found it beneficial and suggested that such training should be conducted either annually or biannually.

## Contact with pharmaceutical sales representatives

Contact with pharmaceutical sales representatives (PSRs) happened once a week for 60.00% of the prescribers interviewed and twice a week for 30.00%. Five prescribers (50.00%) mentioned that they spend less than or equal to 10 minutes with PSRs, four prescribers (40.00%) reported that they spent 11–20 minutes with PSRs, and one prescriber (10.00%) spent more than or equal to 21 minutes with PSRs. Most of the prescribers had these interactions in their consulting rooms (Fig 1).

**Table 8. Demographic characteristics of prescribers.**

| Variable | Number | Percentage |
| --- | --- | --- |
| **Sex** | | |
| Male | 3 | 30.00 |
| Female | 7 | 70.00 |
| **Age** | | |
| 25–30 | 4 | 40.00 |
| 31–35 | 5 | 50.00 |
| 36–40 | 1 | 10.00 |
| **Profession/Rank** | | |
| Medical officer | 9 | 90.00 |
| Specialist | 1 | 10.00 |
| **Work experience (years)** | | |
| 1–5 | 6 | 60.00 |
| 6–10 | 4 | 40.00 |

### Work-related factors

Only one prescriber (10.0%) reported having access to a Standard Treatment Guideline (STG) or Essential Medicines List (EML) in the consulting room. Half of the prescribers (5, 50.0%) indicated that they refer to clinical guidelines, with three doing so weekly and one monthly. The majority (70%) often rely on internet searches when prescribing a new medication (Table 9). Regarding the Drugs and Therapeutics Committee (DTC) at UGMC, 80% of prescribers were unsure of its existence, while 20% confirmed that a functional DTC is in place. Additionally, 30% of prescribers strongly agreed that DTCs play a critical role in promoting adherence to rational use of medicines (RUM), and that pharmaceutical representatives are a valuable source of information on new drugs. All prescribers cited at least one reason for potential non-adherence to RUM standards. The most frequently mentioned factors were the complexity of diseases (8, 80%) and patient demands (8, 80%) (Fig 2).

### Discussion

The following indicators, "percentage of encounters with an injectable prescribed", "average quantity of medications per encounter", and "percentage of encounters with antibiotics" met the WHO reference standards. On the other hand, the indicators, "degree of documentation of diagnosis", "proportion of prescriptions with generic names", and "proportion of prescriptions from EML" failed to meet the standard. Rational drug prescribing (IRDP) had an overall observed index of 4.58, which was less than the optimum index of 6 suggesting a low rate of rational drug prescribing practices.

The average number of medications per patient encounter was 1.4 (SD = 1.61). This value is within the WHO-recommended range of less than 2. Again, the study result was much lower than similar research findings in the United Arab Emirates (UAE) at 4.9 [17], Uganda and India at 3.2 [18,19], Ghana at 3 [10], Nigeria at 2.76 [20], Eritrea at 1.78 [21] and Ethiopia at 1.69 [22]. This finding could primarily be as a result of the low proportion of patients with comorbidities seen in this study (7.17%) thus reducing the need for multiple medications. In many LMICs, polypharmacy is a prevalent issue which, if left unchecked, could result in negative consequences including the high risk of negative drug-drug and drug-food interactions in the patient, and ultimately increased treatment costs.

The World Health Organization recommends that only generic names be used in prescriptions by healthcare institutions. There is a strong emphasis on the importance of prescribing medications using their generic names for patient safety, since this ensures that the drugs are identified and enhances communication between pharmacists and prescribers, among other advantages [8]. To encourage the safe use of pharmaceuticals, avoid confusion during the dispensing

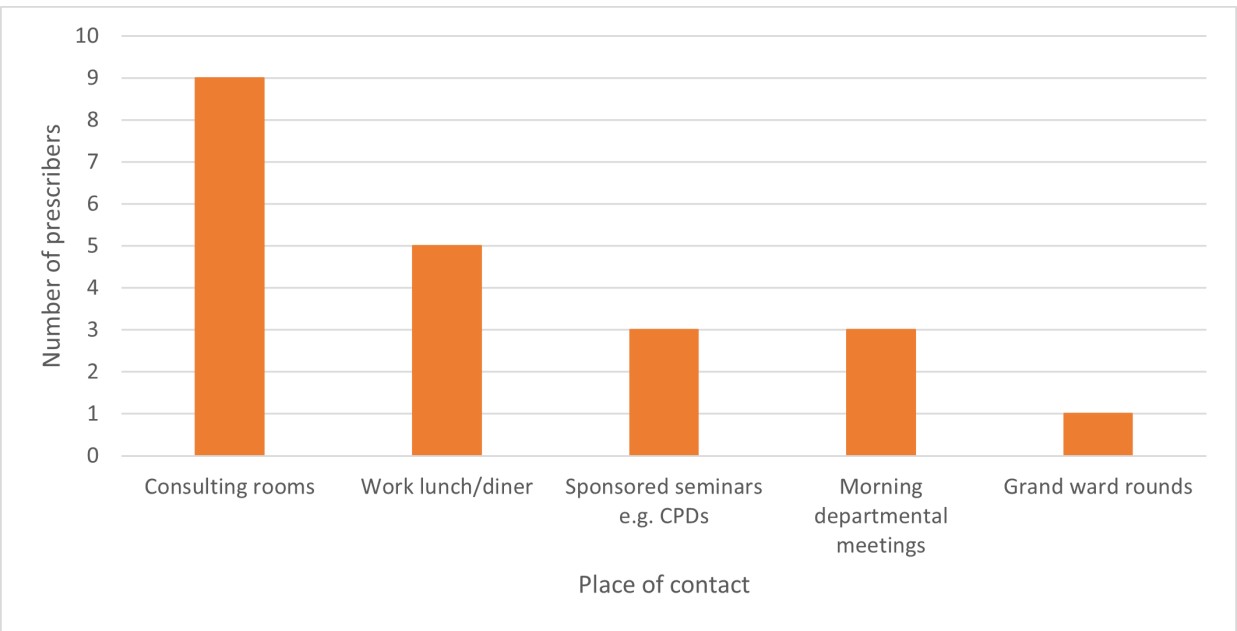

**Fig 1. Places where prescribers interact with medical sales representatives.**

**Table 9. Sources of information prescribers rely on when prescribing.**

| Source of information | Frequency of relying on a source of information for prescribing | | | | |
|---|---|---|---|---|---|
| | Almost always | Often | Sometimes | Rarely | Almost never |
| Standard Treatment Guideline | 0 (0.00) | 2 (20.00) | 6 (60.00) | 1 (10.00) | 1 (10.00) |
| Essential Medicine List | 0 (0.00) | 1 (10.00) | 1 (10.00) | 6 (60.00) | 2 (20.00) |
| Pharmaceutical Sales Representatives | 0 (0.00) | 2 (20.00) | 1 (10.00) | 4 (40.00) | 3 (30.00) |
| Pharmaceutical brochures | 0 (0.00) | 0 (0.00) | 3 (30.00) | 5 (50.00) | 2 (20.00) |
| Internet searches | 0 (0.00) | 7 (70.00) | 3 (30.00) | 0 (0.00) | 0 (0.00) |
| Medical apps, e.g., Medscape | 2 (20.00) | 6 (60.00) | 1 (10.00) | 1 (10.00) | 0 (0.00) |
| Journal articles | 0 (0.00) | 3 (30.00) | 5 (50.00) | 1 (10.00) | 1 (10.00) |
| Advertisements | 0 (0.00) | 0 (0.00) | 3 (30.00) | 2 (20.00) | 5 (50.00) |
| Others | 0 (0.00) | 1 (10.00) | 2 (20.00) | 3 (30.00) | 4 (40.00) |

process, and lower the overall cost of filling name-brand prescriptions, generic prescribing is essential [23]. The study found that 34.88% (observed IRDP = 0.35) of prescriptions were written in their generic names which was below the standard (optimum IRDP = 1). This finding was lower than that of comparable studies conducted in Ghana (63.8%) [24], Sierra Leone (57%) [25], and Lusaka (56.1%) [26]. A possible justification for this could be that the electronic health records system in use at the hospital identifies medications largely by their brand names. As such, prescribers are more likely to select brand names out of convenience to generate prescriptions electronically. Promotional activities by PSRs and prescribers' perceptions that certain branded medications are superior in terms of efficacy may also be contributing factors to the low rate of generic prescribing in the facility. Feedback from the questionnaires administered found that 60% of prescribers met with pharmaceutical sales representatives once a week and that half of them spend no more than ten minutes in these meetings, which typically happen in their consulting rooms. A streamlined drug database, as well as regulating interactions between PSR's and prescribers could significantly improve generic prescribing practices.

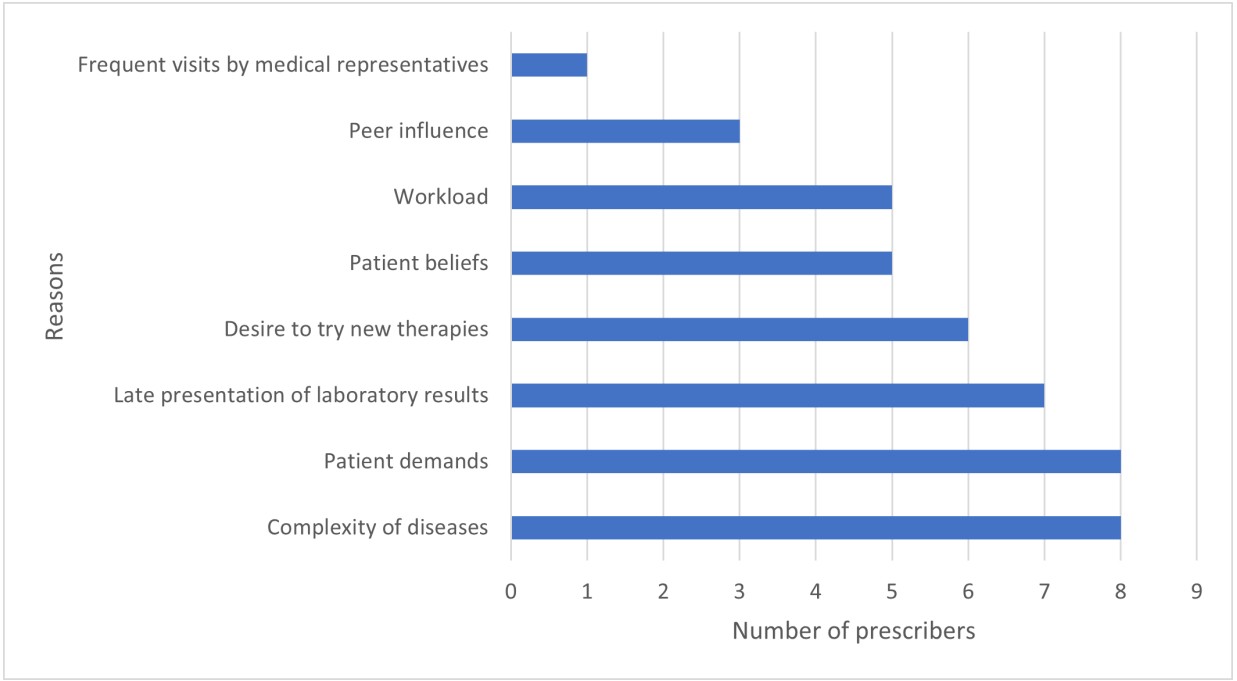

**Fig 2. Reasons why prescribers may be nonadherent to RUM standards.**

The WHO recommends that all visits have a recorded diagnosis, however, the percentage of diagnoses that were recorded was 50.83%. It is imperative for all prescriptions to have a diagnosis on them to help the pharmacists link the patient's illness to the drugs prescribed in order to provide appropriate pharmaceutical counseling to patients during drug dispensing. The percentage obtained from this study indicates a low rate of diagnosis documentation which could be accounted for by the time constraint of such a busy settings, where physicians prioritize treatment over documentation [27]. Additionally, diagnostic uncertainty and the fact that the EMR system does not require a documented diagnosis for each prescription could also be contributing factors. The IRDP for the indicator "recording of diagnosis" was 0.91, although close to the optimum value of 1, is still unsatisfactory. This calls for the sensitization of prescribers on the essence of recording diagnoses as a component of rational prescribing in the family medicine clinic at UGMC. This may be done through regular engagements with prescribers by pharmacists at clinical meetings and through recommendations when prescriptions are sent to the pharmacy.

Each country is expected to design its essential medicines list (EML) based on the health needs of the populace and following WHO standards. The WHO also requires all medications prescribed in health facilities to be found in the EML [28]. This ensures that patients receive safe, effective, and good-quality medications while reducing the risks associated with taking counterfeit medications. From the study, it was observed that 72.38% of medicines prescribed were in the EML of Ghana, which is lower than the WHO recommendation of 100%. The index of 0.72 which is far lower than the optimal value [1] is also a mark of poor rational prescribing. This finding was also lower than the findings of a similar study by WHO (89.0%) [9]. This might be due to the non-availability of medication formularies to all prescribers owing to the inactivity of the Drugs and Therapeutics Committee (DTC) during the data collection period. The DTC is responsible for developing the hospital formulary and essential medicines list [29]. Furthermore, feedback from the prescribers suggested that only 1 out of 10 had an EML available in the consulting room. A small number of prescribers admitted to knowing about the presence of a DTC despite there being one in the facility. Additionally, 60% of the prescribers agreed that the

DTC is important to ensure RUM. Management must therefore ensure that the DTC is reinstated and well-resourced to execute its functions to help ensure the rational use of medicines. The survey also revealed that aside from the Standard Treatment Guidelines, most prescribers also used the British National Formulary (BNF), the internet, and information from medical representatives as references for drug selection (Table 9). This reliance on other reference materials is mainly because of the outdated nature of the national EML of Ghana as it was last updated in 2017 [30]. UGMC is a specialist hospital where many novel medications are used. As a result, these medications would most likely not be captured in the current EML. This further highlights the need for a uniform and regularly updated EML to standardize prescribing practices.

The percentage of encounters with prescribed antibiotics was found to be much lower than the WHO suggested value of fewer than 30 percent, at 12.74%. Comparable studies conducted in Ghana reported percentages of 11.9% [24,31] and 6.5%, showcasing a trend of adherence to the recommended value. In contrast, variations were observed in studies conducted in India (22%) [32] and Uganda (56%) [16], where antibiotic prescription rates were higher. These numbers were notably much lower than those seen in studies carried out in Sudan (63%) [33] and some regions in Africa (45.9%) [16]. High percentages of antibiotic prescriptions could be the result of several factors, including higher infection rates, changing resistance patterns and increasing prevalence of multidrug-resistant organisms [34]. UGMC has an Antimicrobial Stewardship (AMS) committee whose core mandate is to ensure that antibiotics are used appropriately. Therefore, the low prescription of antibiotics seen in this study may be a reflection of the effectiveness of the interventions made by the committee. Misuse of antibiotics is a major driver of antimicrobial resistance [34] which leads to prolonged hospital stays and increased costs to the patient. AMS activities, therefore play an important role in curbing this practice.

The percentage of injections prescribed was 4.17% with an optimal index of 1. This falls within the WHO recommendation of less than 25% and was much lower than findings from studies carried out in some regions in Africa (25%) [9], Pakistan (27.1%) [9] and Tanzania 18.1% [35]. However, it was greater than the results of a different study conducted in Saudi Arabia, with a percentage of 2% [8]. The observed findings for this indicator are consistent with the clinical context of the family medicine clinic, which primarily serves ambulatory patients requiring acute care. This department also operates in conjunction with an active emergency department where patients who would require injectables are taken care of. Inappropriate use of injections puts patients at risk of developing pain in the muscle site and increases their risk of contracting diseases such as HIV and hepatitis through contaminated equipment as well as toxicity due to their high bioavailability [16].

The number of drugs in a prescription was found to be significantly associated with the presence of a comorbidity in a patient. Patients with comorbidities had a 52.2% lower prevalence rate of polypharmacy compared to those without comorbidities. This unexpected result was less than a similar study done in Ethiopia, where prescriptions for patients with comorbid condition had a five-fold higher likelihood of having polypharmacy than those without comorbidities [4]. Furthermore, a study conducted in Kenya revealed that prescriptions for patients with comorbid diseases had a 6.3-fold higher likelihood of exhibiting polypharmacy in comparison to those without comorbidities [36]. This finding is expected, as the management of comorbidities often necessitates the use of multiple medications to address the diverse clinical needs of patients. However, it may also highlight a limitation in the data collection process, which focused solely on newly issued prescriptions and did not account for medications previously prescribed and still in use. Patients on chronic therapy may have existing medication supplies and therefore may not require new prescriptions at the time of the study. Furthermore, a significant proportion of the physicians interviewed reported prior training on Rational Use of Medicines (RUM), which may have positively influenced their prescribing patterns. These observations underscore the importance of institutionalizing regular RUM training for prescribers, alongside implementing policy-driven interventions to promote more consistent and rational prescribing practices.

There was a significant correlation between sex and the prescription of antibiotics, which is consistent with other research findings [37].

In this study, it was projected that female patients would receive 0.536 times as many antibiotic prescriptions as male patients. Similarly, in the UK, a study conducted between 2013 and 2015 revealed that the majority (62.6%) of antibiotic prescriptions were for females as well [37]. These findings are substantially different from a US study that found that patients who got antibiotics were more likely to be male (61%) [38]. The observed difference can be attributed to changes in the occurrence of specific infectious illnesses, like urinary tract infections (UTIs), which are more common in adult females than in adult males [39]. Studies also show that men do not visit the hospital as often as women, which may significantly contribute to the difference in antibiotic prescribing between genders [40].

The CDC defines chronic diseases as conditions lasting a year or more, needing constant medical attention, and possibly restricting daily activities [41]. According to this study, individuals with chronic diseases were expected to receive nearly 2 times the amount of antibiotics given in comparison to those without chronic conditions. A Kenyan study reported similar results, where prescriptions for chronic illnesses had eight times more cases of polypharmacy than those for non-chronic illnesses [36]. This might be because chronic conditions, just like comorbidities, often require multiple drugs for effective management, and symptomatic treatment may lead to additional drug prescriptions [4]. Again, the presence of chronic conditions was found to be significantly associated ($p = 0.048$) with the number of drugs prescribed from the EML in this study. For a patient with a chronic ailment, the expected number of prescriptions from the EML is 0.256 times higher than those without a chronic condition. This association introduces a new perspective on the therapeutic management of patients by prescribers. It implies that clinical decisions must be supported by guidelines to provide patients with a cost-effective treatment regimen, since patients with chronic illnesses are most likely going to be on multiple medications. Notably, prescriber feedback identified two key contributing factors to non-adherence: the complexity of diseases (80%) and patient demands (80%). These results highlight the challenges that prescribers may have in the management of complex medical conditions and the influence that patient expectations have on prescribing practices. Prescribers may be more cautious in prescribing if they receive regular training on RUM. A majority of them expressed this. They believed that such regular training intervals would significantly contribute to improving adherence to RUM standards. The strengths of this study include a focus on a clinical area that could provide insights into both primary and secondary care prescribing behaviors. Additionally, the combination of quantitative data with qualitative interviews provided context and deeper insights into prescribing behaviors. On the other hand, a focus on just one institution as well as a small sample size relative to the overall population may limit the generalizability and representativeness of the findings from this study.

## Conclusions

The study findings show that among the six prescribing indicators, the average number of medicines per prescription, percentage of prescriptions with injectables, and the percentage of prescriptions with antibiotics all conformed to the WHO reference standards. Patient-related factors that affect prescription patterns are; the presence of a chronic condition and sex. Patients having a chronic condition significantly affected the total number of drugs prescribed, the presence of an antibiotic in the prescription and the number of drugs prescribed from the EML. Also, females were less likely to be prescribed an antibiotic compared to males. Additionally, upon engaging with prescribers, it became evident that while a majority were familiar with the concept of rational use of medicines (RUM), only a few had received formal training on this subject. Furthermore, prescribers highlighted that the primary reasons for non-adherence to RUM standards were the complexity of diseases and patient demands. We recommend that rational drug prescribing be enhanced through training, guidelines, EML distribution, drug and therapeutics committee support and integrated Clinical Decision Support Systems (CDSS). Additionally, enforcing DTCs in healthcare institutions, and expanding research to include patient and healthcare provider perspectives would also greatly improve rational use of medicines.

## Author contributions

**Conceptualization:** Nana Ama Buadiba Osei, Kofi Bedu Tetteh, Bismark Sarfo.

**Data curation:** Nana Ama Buadiba Osei.

**Formal analysis:** Nana Ama Buadiba Osei.

**Investigation:** Nana Ama Buadiba Osei.

**Methodology:** Nana Ama Buadiba Osei.

**Project administration:** Nana Ama Buadiba Osei, Daniel Freeman Owusu Ansah, Kofi Bedu Tetteh, Angela Appiah Kuffuor, Bismark Sarfo.

**Resources:** Nana Ama Buadiba Osei, Daniel Freeman Owusu Ansah, Kofi Bedu Tetteh, Angela Appiah Kuffuor.

**Software:** Nana Ama Buadiba Osei.

**Supervision:** Angela Appiah Kuffuor, Fareeda Serwaa Brobbey, Bismark Sarfo.

**Validation:** Nana Ama Buadiba Osei, Kofi Bedu Tetteh, Bismark Sarfo.

**Visualization:** Nana Ama Buadiba Osei.

**Writing – original draft:** Nana Ama Buadiba Osei, Daniel Freeman Owusu Ansah.

**Writing – review & editing:** Daniel Freeman Owusu Ansah.

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
