## [Decision Letter · Decision Letter 0]

10 Oct 2024

Dear Dr. Osei,

Thank you for submitting your manuscript to PLOS ONE. After careful consideration, we feel that it has merit but does not fully meet PLOS ONE’s publication criteria as it currently stands. Therefore, we invite you to submit a revised version of the manuscript that addresses the points raised during the review process.

Please do pay careful attention to the queries raised by the review on methodological considerations, and on additional information requested in the discussion. Please also ensure that your manuscript fulfills all the requirements for publication in this journal.

We look forward to receiving your revised manuscript.

Kind regards,

Fatima Suleman, PhD

Academic Editor

PLOS ONE

Journal Requirements:

4. We note you have included a table to which you do not refer in the text of your manuscript. Please ensure that you refer to Table 10 in your text; if accepted, production will need this reference to link the reader to the Table.

Reviewers' comments:

Reviewer's Responses to Questions

**Comments to the Author**

1. Is the manuscript technically sound, and do the data support the conclusions?

Reviewer #1: Yes

Reviewer #2: Yes

2. Has the statistical analysis been performed appropriately and rigorously?

Reviewer #1: Yes

Reviewer #2: Yes

3. Have the authors made all data underlying the findings in their manuscript fully available?

Reviewer #1: Yes

Reviewer #2: Yes

4. Is the manuscript presented in an intelligible fashion and written in standard English?

Reviewer #1: Yes

Reviewer #2: Yes

Reviewer #1: A good study appropriately written

Some comments are listed below:

1. Study design - Is it a cross sectional design? Prescription records of the past have been studied, so it should be a retrospective record based design

2. Random selection of the samples need to be elaborated. 50 prescriptions every monthfor17 months adds up to 850 prescriptions, but 600 only have been studied. How was this reduction made?

3. Is the IRDP a validated index? The WHO recommended optimal values for some of the indicators are not absolute numbers but 'below a cut off value". Then how the index can be calculated considering the existence of a wide range of numbers below the cut off?

4. Table 3 could be avoided by just adding one more row to Table 2 with 'Overall IRDP" and mention the optimal IRDP in text.

5. In results, many tables with regression analysis for association of demographic factors has been reported. But these findings are not reflected in the conclusion.

6. Though this is a quaternary facility, there is no mention about the availability of the prescribed drugs in the facility, is it a government or private facility?

Reviewer #2: The manuscript is well written. However, my only recommendation would be to include a few lines in the Discussion on reasons for less than optimum values for some prescribing indicators, eg. generic prescribing and prescribing from the EML and/or hosrpiatal formulary, and also what measures were taken or planned to be taken to improve these.

**Do you want your identity to be public for this peer review?** For information about this choice, including consent withdrawal, please see our Privacy Policy

Reviewer #1: No

Reviewer #2: **Yes:** Dr. Ratinder Jhaj

---

## [Author Response · Author response to Decision Letter 1]

20 Nov 2024

Response to Reviewer #1

1. Comment 1: Study design - Is it a cross-sectional design? Prescription records of the past have been studied, so it should be a retrospective record-based design.

a) Response 1: I agree that the study design requires further clarification. The study involved two components: a retrospective review of prescription data from patient medical records to assess WHO core prescribing indicators at UGMC from January 2022 to May 2023, and interviews with prescribers conducted during the data collection period in August 2023. These interviews aimed to gather additional insights into prescribing practices. The study design has been updated to reflect this dual approach in the "Methods" section.

2. Comment 2: Random selection of the samples needs to be elaborated. 50 prescriptions every month for 17 months adds up to 850 prescriptions, but 600 only have been studied. How was this reduction made?

b) Response 2: I acknowledge the need for clarity. The WHO guideline that this study followed recommends at least 600 encounters for such studies (Ofori-Asenso, 2016; World Health Organization, 1993). Following the extraction of the 23,690 prescriptions, the data was exported to the STATA software where a command was issued to randomly select 600 prescriptions systematically from the total number of prescriptions. This helped ensure manageable data analysis while maintaining representativeness. Initially, the approach involved selecting 50 prescriptions monthly but this was later modified. Its inclusion in the write-up was a typographical error, which has been corrected in the revised manuscript.

3. Comment 3: Is the IRDP a validated index? The WHO recommended optimal values for some indicators are not absolute numbers but "below a cut-off value." How can the index be calculated considering the existence of a wide range of numbers below the cut-off?

c) Response 3: The IRDP was adapted from established methodologies used in similar studies and is rooted in the mathematical model developed by Zhang and Zhi (1995), which has been widely utilized to evaluate rational drug use (RUM). This model assigns an optimal index of 1 for each prescribing indicator, where values closer to 1 indicate more rational prescribing. While the WHO-recommended values for some indicators are presented as thresholds (e.g., "below a cut-off value"), the IRDP provides a standardized approach to quantify deviations from these recommendations, enabling consistent comparisons.

4. Comment 4: Table 3 could be avoided by just adding one more row to Table 2 with 'Overall IRDP' and mentioning the optimal IRDP in the text.

d) Response 4: Table 3 has been removed, and the "Overall IRDP" has been added as a row in Table 2. The optimal IRDP has been mentioned in the text for clarity.

5. Comment 5: In the results, many tables with regression analysis for the association of demographic factors have been reported. However, these findings are not reflected in the conclusion.

e) Response 5: The conclusions have been revised to incorporate the key findings of the regression analyses, particularly the associations between demographic factors and prescribing patterns.

6. Comment 6: Though this is a quaternary facility, there is no mention of the availability of the prescribed drugs in the facility. Is it a government or private facility?

f) Response 6: Thank you for highlighting this omission. The manuscript now specifies that the study was conducted in a quaternary, government-funded facility, and the availability of prescribed drugs has been briefly discussed in the "Discussion" section.

Response to Reviewer #2

7. Comment: Include a few lines in the Discussion on reasons for less than optimum values for some prescribing indicators (e.g., generic prescribing and prescribing from the EML or hospital formulary) and what measures were taken or planned to be taken to improve these.

g) Response: I have expanded the "Discussion" section to explore reasons for less-than-optimal values under generic prescribing and prescribing from the EML including the inactivity of the Drugs and Therapeutics Committee as well as the absence of a medicines formulary in the facility. I have also indicated some measures that can be taken to improve these values including the reinstatement of the DTC and providing them with the resources needed to work as well as the development and dissemination of a hospital formulary to prescribers.

---

## [Decision Letter · Decision Letter 1]

17 Dec 2024

Dear Dr. Osei,

Thank you for submitting your manuscript to PLOS ONE. After careful consideration, we feel that it has merit but does not fully meet PLOS ONE’s publication criteria as it currently stands. Therefore, we invite you to submit a revised version of the manuscript that addresses the points raised during the review process.

We look forward to receiving your revised manuscript.

Kind regards,

David Adedia, Ph.D

Academic Editor

PLOS ONE

Additional Editor Comments:

The authors should rewrite the Abstract by writing a conclusion with some recommendations.

The demographic predictors are very few and additional ones would have improved the results.

Reviewers' comments:

Reviewer's Responses to Questions

**Comments to the Author**

Reviewer #1: All comments have been addressed

Reviewer #2: All comments have been addressed

Reviewer #3: (No Response)

Reviewer #4: (No Response)

2. Is the manuscript technically sound, and do the data support the conclusions?

Reviewer #1: Yes

Reviewer #2: Yes

Reviewer #3: No

Reviewer #4: Partly

3. Has the statistical analysis been performed appropriately and rigorously?

Reviewer #1: Yes

Reviewer #2: Yes

Reviewer #3: No

Reviewer #4: Yes

4. Have the authors made all data underlying the findings in their manuscript fully available?

Reviewer #1: Yes

Reviewer #2: Yes

Reviewer #3: Yes

Reviewer #4: Yes

5. Is the manuscript presented in an intelligible fashion and written in standard English?

Reviewer #1: Yes

Reviewer #2: Yes

Reviewer #3: Yes

Reviewer #4: No

Reviewer #1: The authors have addressed all the queries

In reply to query 2, it is mentioned that there was typographical error with respect to random sampling description. It appears that it is more than just a typographical error. This seems unacceptable when submitting to reputed journals like PLOS ONE, and needs to be avoided.

Other queries have been addressed satisfactorily.

Reviewer #2: All suggestions/comments from previous review have been addressed. The manuscript requires no further modifications from my side

Reviewer #3: Abstract

Any reason why patients at the out-patient pharmacy were being prescribed injections would be very much appreciated.

Introduction

Your citation of studies is not correct. For example, Mahali (2012) and Baba and Salifu (2019) are totally out of place. There is another one on page 6, line 128.

Results

Table 1 is to scanty. I was expecting to see more variables here. For example, you could show a categorical variable of number of drugs per prescription, that is < 1 (0), 1-2, 3-4, >4, and so on.

If you have 23,690 records and the minimum required is 600, why did you not go for more to be extra sure of the sure of the study power?

Page 7, line 148, you mentioned that patient identities were either removed of anonymized. You need to be consistent here. What was actually done to patient identities?

Kindly explain the rationale behind the use of zero-inflated Poisson regression in your study. What is the relationship between the mean number of drugs prescribed in this study and the variance?

Results

What is the meaning of “mean age of the patient was 43.82+17.88 years? The mean normally go with the standard deviation. Why do you have inter-quartile range (IQR) here? The IQR normally goes with the median. These are not on Table 1, however you have referred to Table 1. Why is that so?

It will be better to have “observed values” and “optimal values” for Table 2 to help reviewers calculate the index value as specified in your methods.

The p-values for Age as indicated in both Tables 5 and 6 look too high compared to the 95% confidence intervals indicated. Kindly check your statistical values again.

Are you IRR values coefficients or real rate ratios?

Discussion

On page 21, line number 402, you stated that “for those with comorbidities the number of drugs was 0.346 times higher… Are you saying that those with comorbidities had less drugs compared to those without comorbidities? There is a similar statement on line 412.

On page 9 you stated that patients with comorbidities have a 52.2% lower prevalence rate of the total number of medicines prescribed compared to those without comorbidities. However, your abstract says that patients without comorbidities were 65.4% less likely to have polypharmacy. These are quite confusing. It well established that patients with comorbidities have more prescribed medicines because more diseases that are not related may be treated.

You need to look at your discussion again and explain your statistical analysis well. These seem to be a few discrepancies here and there.

Reviewer #4: In this manuscript, the authors aimed to assess the drug prescription patterns of the family medicine clinic at the Outpatient Pharmacy of the University of Ghana Medical Centre using the WHO core prescribing indicators. The study is interesting and provides an important information to the healthcare system. However, a number of minor and major comments need to be addressed before its publication.

Abstract

1. Provide adjusted odds ratio along with 95% CI for significant independent variables.

2. I suggest the authors to include a concise recommendation in the conclusion section.

Methods and results

1. The study design is cross-sectional. I suggest the authors to omit ‘retrospective’ in the study design. Simply indicate that data were collected retrospectively in the ‘Data collection tool and approach’ section. This is because retrospective study design is one of the longitudinal study designs in which the dependent variable and independent variables are measured at different points in time.

2. Massive English editing required. Coherence and readability of the manuscript need improvement.

3. Clearly state the selection criteria of the study setting over other clinics in Ghana.

4. For ease of understanding and readability, I suggest the authors to segregate the method section into sub-sections such as study design and setting, target population, sample size determination, data collection tool and approach, variables, data quality control, ethical consideration, and statistical analysis.

5. Provide an operational definition for polypharmacy and rational medicine use.

6. Did you conduct a face and content validity, and pre-test for the questionnaire that was used in interviewing the prescribers?

7. Citation for index of rational drug prescribing is incorrect [Page 6; Line 128].

8. Did your study conforms to the principles outlined in the Declaration of Helsinki?

9. State the optimal values for the WHO prescribing indicators in a table format.

10. How was the 600 prescriptions selected from the total 23,690?

11. Explain in detail the statistical analysis used in this study.

12. What is your inclusion criteria and sampling technique utilized for recruiting the prescribers?

13. The knowledge and training of prescribers towards rational use of medicines, contact of prescribers with pharmaceutical sales representatives, and work-related factors are not in line with the objectives of the study. It cannot be representative to the views of the ordinary prescribers.

14. Explain in detail the number of items included in the questionnaire. State the number of open-ended and closed-ended questions.

Discussion

1. Reason in the difference of results from the current study and other similar studies were not stated. Moreover, provide policy implications for the various findings in your study.

2. Explain in detail why antibiotic prescription in Ghana was consistent with the WHO standard. Such sort of information will provide an insight and lesson to countries with similar socio-economic status.

3. Avoid detail figures (for instance 95% CI) in the discussion section and take more emphasis on interpreting the results while providing explanations and policy implications for the findings.

4. State the limitations of the study.

5. Provide recommendations along with the conclusion of the study.

**Do you want your identity to be public for this peer review?** For information about this choice, including consent withdrawal, please see our Privacy Policy

Reviewer #1: No

Reviewer #2: **Yes:** Dr. Ratinder Jhaj

Reviewer #3: No

Reviewer #4: No

---

## [Author Response · Author response to Decision Letter 2]

31 Jan 2025

Response to Reviewer #3

1. Comment 1: Any reason why patients at the out-patient pharmacy were being prescribed injections would be very much appreciated.

a) Response 1: The Out Patient department has a 24-hour detention area for patient observation, such patients are administered stat doses of intravenous medications and are subsequently discharged or admitted based on the physician’s assessment.

2. Comment 2: Your citation of studies is not correct. For example, Mahali (2012) and Baba and Salifu (2019) are totally out of place. There is another one on page 6, line 128.

b) Response 2: References have been corrected accordingly

3. Comment 3: Table 1 is too scanty. I was expecting to see more variables here. For example, you could show a categorical variable of number of drugs per prescription, that is < 1 (0), 1-2, 3-4, >4, and so on.

c) Response 3: The data obtained was analyzed in direct relation to the objectives of study which focused on the average number of drugs prescribed per visit not per prescription

4. Comment 4: If you have 23,690 records and the minimum required is 600, why did you not go for more to be extra sure of the sure of the study power?

d) Response 4: Owing to a lack of resources and time constraints the number of records assessed was limited to the minimum required by the WHO. A larger sample size will be employed in follow-up studies

5. Comment 5: Page 7, line 148, you mentioned that patient identities were either removed of anonymized. You need to be consistent here. What was actually done to patient identities?

e) Response 5: Patient identifiers were anonymized during the extraction process and before analysis, they were deleted completely. This will be clearly indicated in the manuscript.

6. Comment 6: Kindly explain the rationale behind the use of zero-inflated Poisson regression in your study. What is the relationship between the mean number of drugs prescribed in this study and the variance?

f) Response 6: The zero-inflated Poisson regression was used for the “number of injectables” and “antibiotics found on a prescription”. A prescription can have at least one medication but this medication may neither be an injectable nor an antibiotic. This increased the number of zeros (0) data points found under these variables hence the use of the zero-inflated Poisson regression. The variance was not reported on in this study with respect to the number of drugs prescribed

7. Comment 7: What is the meaning of “mean age of the patient was 43.82+17.88 years? The mean normally goes with the standard deviation. Why do you have inter-quartile range (IQR) here? The IQR normally goes with the median. These are not on Table 1, however you have referred to Table 1. Why is that so?

Response 7: This statement should read “mean age of the patient was 43.82(SD =17.88 years) and the inter-quartile range has been taken out. The reference to table one was for all other statements made in that paragraph. Reference has been shifted up to reflect that

8. Comment 8: It will be better to have “observed values” and “optimal values” for Table 2 to help reviewers calculate the index value as specified in your methods.

Response 8: The values for observed and optimal are written in the text following the recommendation of one of the reviewers to reduce the number of tables presented.

9. Comment 9: The p-values for Age as indicated in both Tables 5 and 6 look too high compared to the 95% confidence intervals indicated. Kindly check your statistical values again.

Response 9: Below is the direct output from stata after the raw data was analyzed.

10. Comment 10: Are you IRR values coefficients or real rate ratios?

Response 10: They are the coefficients

11. Comment 11: On page 21, line number 402, you stated that “for those with comorbidities the number of drugs was 0.346 times higher… Are you saying that those with comorbidities had less drugs compared to those without comorbidities? There is a similar statement on line 412. On page 9 you stated that patients with comorbidities have a 52.2% lower prevalence rate of the total number of medicines prescribed compared to those without comorbidities. However, your abstract says that patients without comorbidities were 65.4% less likely to have polypharmacy. These are quite confusing. It well established that patients with comorbidities have more prescribed medicines because more diseases that are not related may be treated. You need to look at your discussion again and explain your statistical analysis well. These seem to be a few discrepancies here and there.

Response 12: From the analysis done patients with comorbidities have a 52.2% lower prevalence rate of polypharmacy compared to those without comorbidities. The text has been refined to clarify this.

Response to Reviewer #4

1. Comment (Abstract): Provide adjusted odds ratio along with 95% CI for significant independent variables. I suggest the authors to include a concise recommendation in the conclusion section.

Response: Adjusted odds ratios have been included and the following recommendations have been made in the document

2. Comments (Methods and Results)

a. The study design is cross-sectional. I suggest the authors to omit ‘retrospective’ in the study design. Simply indicate that data were collected retrospectively in the ‘Data collection tool and approach’ section. This is because retrospective study design is one of the longitudinal study designs in which the dependent variable and independent variables are measured at different points in time.

Response: Corrections have been made

b. Massive English editing required. Coherence and readability of the manuscript need improvement.

Response: The appropriate corrections have been made

c. Clearly state the selection criteria of the study setting over other clinics in Ghana.

Response: This study was carried out at UGMC; a newly established hospital, to provide baseline information beneficial in identifying quality improvement areas for subsequent interventions to improve prescribing practices.

d. For ease of understanding and readability, I suggest the authors to segregate the method section into sub-sections such as study design and setting, target population, sample size determination, data collection tool and approach, variables, data quality control, ethical consideration, and statistical analysis.

Response: The various subheadings have been included

e. Provide an operational definition for polypharmacy and rational medicine use.

Response: Rational medicine use encompasses all the dependent variables being studied namely; Polypharmacy, Generic name-based prescribing, Antibiotics prescribed, Injections prescribed, Adherence to EML prescribing and polypharmacy is defined as the number of medications prescribed per patient encounter.

f. Did you conduct a face and content validity, and pre-test for the questionnaire that was used in interviewing the prescribers?

Response: The questionnaire was pre-tested among prescribers from the anesthesia, internal medicine and surgery outpatient clinics at UGMC and then reviewed where necessary based on the responses received.

g. Citation for index of rational drug prescribing is incorrect [Page 6; Line 128].

Response: Citation has been modified according to PLOS requirements

h. Did your study conform to the principles outlined in the Declaration of Helsinki?

Response: Yes

i. State the optimal values for the WHO prescribing indicators in a table format.

Response: The values for observed and optimal are written in the text following the recommendation of one of the reviewers to reduce the number of tables presented and present the optimal values as presented in the manuscript

j. How was the 600 prescriptions selected from the total 23,690?

Response: The WHO guideline that this study followed recommends at least 600 encounters for such studies (Ofori-Asenso, 2016; World Health Organization, 1993). Following the extraction of the 23,690 prescriptions, the data was exported to the STATA software where a command was issued to randomly select 600 prescriptions systematically from the total number of prescriptions. This helped ensure manageable data analysis while maintaining representativeness.

k. Explain in detail the statistical analysis used in this study.

Response: The dependent variables were all count variables thus the use of the Poisson regression model to establish an association. However, the zero-inflated Poisson regression was used for the number of injectables or antibiotics found on a prescription. A prescription can have at least one medication but this medication may neither be an injectable nor an antibiotic. This increased the number zeros (0) values found under these variables hence the use of the zero-inflated Poisson regression. Logistic regression was carried out to establish associations between dependent variables that were categorical variables such as the association between demographic data and whether there was an antibiotic in the prescription

l. What is your inclusion criteria and sampling technique utilized for recruiting the prescribers?

Response: The inclusion criteria consisted of all physicians who were actively involved in prescribing medications to patients at the family medicine clinic during the period of data collection. All prescribers who were on duty during the study period were sampled due to the small sample size.

m. The knowledge and training of prescribers towards rational use of medicines, contact of prescribers with pharmaceutical sales representatives, and work-related factors are not in line with the objectives of the study. It cannot be representative to the views of the ordinary prescribers.

Response: The database used did not link prescribers to their prescriptions, which was a limitation of the study. This connection would only be possible if the study were conducted in real time. However, through discussions with prescribers, I gained valuable insights into their collective behaviors and the factors influencing their prescribing patterns. Although the prescriber data was not directly linked to individual prescriptions, it provided important context to the study’s outcomes. Future studies could address this limitation by collecting data in real time for a more detailed analysis.

n. Explain in detail the number of items included in the questionnaire. State the number of open-ended and closed-ended questions.

Response: 20 items were included in the questionnaire. Below is an explanation of the questions in the questionnaire.

Section A: Demographic details of Respondents

Sex: Captures the respondent's gender (Closed-ended).

Age: Groups respondents into predefined age ranges (Closed-ended).

Profession: Identifies the respondent’s professional designation, with an option to specify others (Closed-ended with "Other" option).

Years of Practice: Assesses prescribers' experience in years (Closed-ended).

Understanding of Rational Use of Medicines (RUM): Evaluates whether respondents know the definition of RUM (Closed-ended).

Parameters of RUM Standards: Tests knowledge of RUM standards (Closed-ended with an "Any other" option).

Section B: Training on RUM

Training on RUM: Asks if the respondent has undergone training on RUM (Closed-ended).

Last Training Date: Determines when the training occurred (Open-ended).

Training Benefits: Assesses the perceived usefulness of the training (Closed-ended).

Training Frequency: Inquires about the ideal training schedule (Closed-ended with "Other" option).

Section C: Contact with Pharmaceutical Sales Representatives

Frequency of Visits: Captures how often prescribers interact with sales reps (Closed-ended).

Interaction Locations: Identifies where these interactions occur (Closed-ended).

Interaction Duration: Estimates the typical duration of interactions (Closed-ended).

Section D: Work-Related Factors That Affect the Rational Use of Medicines

Guideline Availability: Inquires if prescribers have access to guidelines (Closed-ended).

Guideline Reference Frequency: Determines how often guidelines are used (Closed-ended).

Information Sources: Assesses reliance on various information sources for prescribing decisions (Closed-ended).

Functional Drug and Therapeutics Committee: Verifies the existence of a relevant committee at the facility (Closed-ended).

Perception of Representatives and Committees: Seeks agreement on the value of pharmaceutical representatives and therapeutic committees using a Likert scale (Closed-ended).

Section E: Reasons for Non-Adherence to RUM

Factors for Non-Adherence: Explores reasons for not fully adhering to RUM standards (Closed-ended).

Total Open-Ended Questions: 2

Total Closed-Ended Questions: 18.

3. Comments (Discussion)

a. Reason in the difference of results from the current study and other similar studies were not stated. Moreover, provide policy implications for the various findings in your study.

Response: Justification for the variation in findings has been provided

b. Explain in detail why antibiotic prescription in Ghana was consistent with the WHO standard. Such sort of information will provide an insight and lesson to countries with similar socio-economic status.

Response: In UGMC there is an active antimicrobial stewardship committee that develops interventions to regulate the use of antibiotics according to WHO standards. This has been stated in the discussion.

c. Avoid detail figures (for instance 95% CI) in the discussion section and take more emphasis on interpreting the results while providing explanations and policy implications for the findings.

Response: Confidence interval values have been removed from the values in the discussion.

d. State the limitations of the study.

Response: On the other hand, a focus on just one institution as well as a small sample size relative to the overall population may limit the generalizability and representativeness of the findings from this study.

e. Provide recommendations along with the conclusion of the study.

Response: The following recommendations have been added to the conclusion; Rational drug prescribing may be enhanced through training, guidelines, EML distribution, drug and therapeutics committee support and integrated Clinical Decision Support Systems (CDSS). Additionally, enforcing DTCs in healthcare institutions, and expanding research to include patient and healthcare provider perspectives would also greatly improve rational use of medicines.

---

## [Decision Letter · Decision Letter 2]

1 Jun 2025

Thank you for submitting your manuscript to PLOS ONE. After careful consideration, we feel that it has merit but does not fully meet PLOS ONE’s publication criteria as it currently stands. Therefore, we invite you to submit a revised version of the manuscript that addresses the points raised during the review process.

**ACADEMIC EDITOR:**

The authors should proofread the manuscript by using the services of a professional proofreader and provide evidence.

We look forward to receiving your revised manuscript.

Kind regards,

David Adedia, Ph.D

Academic Editor

PLOS ONE

Journal Requirements:

Additional Editor Comments:

A few comments to respond to:

The headings under discussion should be removed. The topic specific discussions could be put in a paragraph.

Reviewers' comments:

Reviewer's Responses to Questions

**Comments to the Author**

Reviewer #3: (No Response)

Reviewer #5: (No Response)

2. Is the manuscript technically sound, and do the data support the conclusions?

Reviewer #3: Yes

Reviewer #5: Partly

3. Has the statistical analysis been performed appropriately and rigorously?

Reviewer #3: Yes

Reviewer #5: No

4. Have the authors made all data underlying the findings in their manuscript fully available?

Reviewer #3: Yes

Reviewer #5: Yes

5. Is the manuscript presented in an intelligible fashion and written in standard English?

Reviewer #3: Yes

Reviewer #5: No

Reviewer #3: 1. The response to comment 3 is not fully acceptable. First of all the average number of drugs per prescription is in itself a benchmark of the WHO instrument. Secondly, a reviewer may request the researcher to add further information for the sake of clarity.

2. Comment 4, researcher mentioned lack of resources and time constraints as reasons for not being able to use more information from the database. It is fine if you decide to use the minimum data size, but lack of resources and time constraints are not good reasons if you want to do a good research. Besides, the research involved secondary data extracted from an existing database.

3. Comment 7, your abstract still has, “The mean number of 33 prescribed medications was 1.4 (n=840)…” instead of mean and standard deviation. There is a very good reason why it is recommended that the standard deviation should be quoted when the mean is indicated. You may rather capture this as “The mean number of 33 (n=840) prescribed medications was 1.4 (SD=xyz)…”.

Reviewer #5: the study needs to be written in more technical language. it lacks consistency in tenses in sentences. the sample size of the prescriber is very less. and analysis on prescription does not provide any new information

**Do you want your identity to be public for this peer review?** For information about this choice, including consent withdrawal, please see our Privacy Policy

Reviewer #3: No

Reviewer #5: No

---

## [Author Response · Author response to Decision Letter 3]

4 Nov 2025

Response to Reviewer #3

1. Comment 1: The response to comment 3 is not fully acceptable. First of all, the average number of drugs per prescription is in itself a benchmark of the WHO instrument. Secondly, a reviewer may request the researcher to add further information for the sake of clarity.

Response: This is well noted. For this study, we wanted to focus on the WHO indicators as this is a baseline study, and future studies will explore in detail the number of prescriptions.

2. Comment 4, the researcher mentioned lack of resources and time constraints as reasons for not being able to use more information from the database. It is fine if you decide to use the minimum data size, but lack of resources and time constraints are not good reasons if you want to do good research. Besides, the research involved secondary data extracted from an existing database.

Response: We acknowledge the disadvantage of a smaller dataset and will rectify that inn future studies.

3. Comment 7, your abstract still has, “The mean number of 33 prescribed medications was 1.4 (n=840)…” instead of mean and standard deviation. There is a very good reason why it is recommended that the standard deviation should be quoted when the mean is indicated. You may rather capture this as “The mean number of 33 (n=840) prescribed medications was 1.4 (SD=xyz)…”.

Response: The standard deviation has been included following the format suggested (Line 33)

Response to Reviewer #5

1. Comment: the study needs to be written in more technical language. it lacks consistency in tenses in sentences. the sample size of the prescriber is very less and analysis on prescription does not provide any new information.

Response: The manuscript has been edited to improve its technical clarity and grammatical consistency. Also, the number of prescribers interviewed reflects the total number of prescribers in that department during the period of the study. Finally, the inability to connect the prescriber interviews to the prescriptions analyzed has been stated as a limitation

---

## [Editor Report · Decision Letter 3]

2 Jan 2026

Assessment of adherence to the World Health Organization’s prescribing indicators at the family medicine clinic of a quaternary facility.

PONE-D-24-26584R3

Dear Dr. Osei,

We’re pleased to inform you that your manuscript has been judged scientifically suitable for publication and will be formally accepted for publication once it meets all outstanding technical requirements.

Kind regards,

Vijayaprakash Suppiah, PhD

Academic Editor

PLOS One

---

## [Editor Report · Acceptance letter]

PONE-D-24-26584R3

PLOS One

Dear Dr. Osei,

I'm pleased to inform you that your manuscript has been deemed suitable for publication in PLOS One. Congratulations! Your manuscript is now being handed over to our production team.

Kind regards,

on behalf of

Dr. Vijayaprakash Suppiah

Academic Editor

PLOS One